# Reproductive Potential and Outcomes in Patients with Hidradenitis Suppurativa: Clinical Profile and Therapeutic Implications

**DOI:** 10.3390/life11040277

**Published:** 2021-03-26

**Authors:** Trinidad Montero-Vilchez, Luis Salvador-Rodriguez, Andrea Rodriguez-Tejero, Manuel Sanchez-Diaz, Salvador Arias-Santiago, Alejandro Molina-Leyva

**Affiliations:** 1Hidradenitis Suppurativa Clinic, Dermatology, Hospital Universitario Virgen de las Nieves, 18012 Granada, Spain; tmonterov@correo.ugr.es (T.M.-V.); l.salvador.rodriguez1991@gmail.com (L.S.-R.); andrea_13_3217@hotmail.com (A.R.-T.); manolo_94_sanchez@hotmail.com (M.S.-D.); alejandromolinaleyva@gmail.com (A.M.-L.); 2Instituto de Investigación Biosanitaria GRANADA, 18012 Granada, Spain; 3Dermatology Department, Faculty of Medicine, University of Granada, 18001 Granada, Spain; 4European Hidradenitis Suppurativa Foundation (EHSF), Dessau-Roßlau, Germany

**Keywords:** hidradenitis suppurativa, acne inversa, pregnancy, reproductive behavior

## Abstract

There are scarce data available regarding the impact of hidradenitis suppurativa (HS) on fertility, course and outcome of pregnancy and risk associated with treatments. The aims of this study are (1) to describe the clinical profile of HS women of childbearing age with and without accomplished reproductive desires and (2) to describe the prescribed treatments based on the fulfillment of reproductive intentions. We conducted a prospective observational study that included 104 HS women of childbearing age, 50.96% (53/104) with unfulfilled reproductive desires. These women were younger (29.08 vs. 42.06 years, *p* < 0.001), less frequently married and higher educated than women with fulfilled reproductive desires. Their age of disease onset was lower, but disease duration was shorter, in concordance with a lower International Hidradenitis Suppurativa Severity Score System (IHS4) and lower number of draining tunnels. Combined oral contraceptives were more frequently prescribed in women with unfulfilled reproductive desires (30.19% vs. 9.80%, *p* = 0.013) while biologics were less used in this group (3.77% vs. 13.73%, *p* = 0.08). In conclusion, a higher educational level and an earlier disease onset, with potential implications in finding a partner, may make the fulfillment of reproductive desires difficult for patients with HS. This study could help clinicians to achieve a better understanding of the specific characteristics of HS during childbearing age and consider reproductive desires when making treatment decisions.

## 1. Introduction

Reproductive health and the fulfillment of genetic desires are an important part of patients’ quality of life that may be affected by hidradenitis suppurativa (HS) [1]. HS is a chronic, recurrent, debilitating inflammatory skin disease of the hair follicle which usually presents after puberty with painful, deep-seated inflamed lesions in the apocrine gland-bearing areas of the body, most commonly the axillae and the inguinal and anogenital regions [2]. It has an estimated prevalence rate of around 1% [3], and it disproportionally affects women of childbearing age [4]. Nevertheless, there are little data available regarding the impact of HS on reproductive desire, pregnancy and the risk associated with treatments [5].

The fertility rate is lower in patients with other inflammatory diseases such as psoriasis [6], rheumatoid arthritis and inflammatory bowel disease [7]. HS is the dermatological disease with the greatest impact on patients’ quality of life [8] and has a similar impact to other non-dermatological conditions such as cardiovascular disease, cancer, diabetes mellitus and chronic obstructive pulmonary disease [9]. HS clinical manifestations cause pain, itching, unpleasant odor and suppuration, among other symptoms, which make life difficult for patients [10] and may have an impact on pregnancy desire. Moreover, HS has a negative effect on sexual function, which is worse in women with active lesions in the groin and genital areas [11]. Poor self-esteem and body image [12,13] and increased risk of anxiety and depression [14,15] may also decrease reproductive desire. Furthermore, HS is associated with comorbidities such as obesity, Crohn’s disease, diabetes type 2 and cardiovascular disease [16], which contribute to a lower fertility rate. The role of the diet is also crucial in fertility rates, as it has been proved that a healthy diet and doing exercise increase the number of natural conceptions [17]. Moreover, the lack of adherence to a Mediterranean diet is associated with more severe HS [18] and polycystic ovary syndrome [19]. High levels of tumor necrosis factor (TNF)-alpha have been associated with both a pathophysiological role in HS [20] and obstetric complications [21], which may explain a decrease in the fertility rate in HS women. In fact, it has been observed that treatment with blockage TNF might be a useful treatment for infertility [21]. 

It also should be considered that some medications commonly prescribed for HS might affect the reproductive function. There are some teratogenic treatments forbidden before and during pregnancy, but there is scarce information on most of them [22]. Clinicians prescribing a drug for women of childbearing age have to consider possible interference with conception, the contraceptive failure caused by medication interactions and the potential risk to mother and fetus caused by the drug if pregnancy occurs [23]. The implications of treatment should be considered in all women with HS who are sexually active, even if they are not actively planning to become pregnant, as approximately half of all pregnancies are unplanned [24] and women usually discover they are pregnant between weeks 4 and 7, a critical window for early fetal development [25].

Thus, the objectives of our study were: (1) to describe the clinical profile of HS women of childbearing age with and without accomplished reproductive desires, (2) to describe the prescribed treatments based on the fulfillment of reproductive desire and the possible implication of treatments on pregnancy and fertility and (3) to describe the clinical profile of HS women receiving first-line biologics based on the fulfillment of reproductive desire.

## 2. Materials and Methods

### 2.1. Design: Prospective Observational Study

Patient selection. This study included all women of childbearing age with a clinical diagnosis of HS who attended the HS Clinic of Hospital Universitario Virgen de las Nieves, Granada, Spain. The diagnosis of HS was performed by a dermatologist relying on clinical findings of (1) typical HS lesions, (2) predilection for intertriginous sites and (3) recurrence [26]. Clinical information was gathered if inclusion and exclusion criteria were met at two different time points: (1) at the first visit to the HS clinic or (2) at the visit where biologic treatment was initiated. A patient’s clinical information could be included in both analyses if biologic treatment was prescribed at the first visit to the HS clinic. The clinical information of a single patient at each time point could also be included in both analyses if the patient was prescribed classic systemic therapy (for example, oral antibiotics) at the first visit to the HS clinic, and afterwards, the patient was prescribed biologic treatment following the recommendations of current treatment guidelines [10]. The flowchart of the participants is shown in Figure 1.


*Inclusion criteria:*
-HS women of childbearing age (15–49 years-old);-Women who were prescribed systemic therapy for the treatment of HS.



*Exclusion criteria:*
-HS women who did not sign the written consent form or young women under 18 whose legal representative did not sign the informed consent;-Climacteric women.


### 2.2. Variables of Interest

#### 2.2.1. Main Variables of Interest

Accomplishment of reproductive desires. Women were asked if they wanted to have a baby now or in the future during the visit, and they were gathered into two groups. If the answer was not, women were classified in the group of women with fulfilled reproductive desires, while if they answered yes, they were included in the group of women with unfulfilled reproductive desires. 

This information was gathered at the first visit to the HS clinic and reassessed by protocol in all women of childbearing age every time a patient was prescribed a systemic treatment. The compatibility of a treatment with pregnancy, fertility and drug washout times was explained to the patient before a drug was prescribed. The patient had to understand and give consent (oral or written as per legal requirements) to the treatment.

#### 2.2.2. Other Variables of Interest

Clinical, sociodemographic and biometric variables were recorded by means of clinical interview and physical examination. Sociodemographic characteristics included sex, age, civil status, level of education, body mass index (BMI), smoking habit, alcohol consumption and family history of HS. Clinical features included age at HS onset, disease duration, Hurley stage, number of affected areas, previous medical and surgical treatments, nodules and abscess and draining tunnel count. 

### 2.3. Ethics

All patients agreed with the treatment regimen and signed a written consent form to use their personal data for the present study. This study was approved by the Ethics Committee of the Hospital Universitario Virgen de las Nieves the 22nd of December 2019, reference number 2390-N-19, and is in accordance with the Helsinki Declaration.

### 2.4. Statistical Analysis

Descriptive statistics were used to evaluate the characteristics of the sample. The Kolmogorov–Smirnov test was used to check the normality of the variables. Continuous data were expressed as mean ± standard deviation (SD) or as the median (25th–75th percentile). The absolute and relative frequency distributions were estimated for qualitative variables. The student’s t-test or the Wilcoxon–Mann–Whitney test were used to compare continuous data (such as age or total abscess and inflammatory nodule (AN) count), and the χ2 test or Fisher’s exact test were applied to nominal data where necessary (such as civil status or Hurley stage). The study size was calculated to determine the proportion of patients with HS and unfulfilled reproductive desires with an alpha error of 5% and a power of 20%; considering a reference population of 450,000 inhabitants, and an estimated prevalence of HS in the general population of 1% [3], the estimated sample size was 49 patients. Significance was set for all tests at two tails, *p* < 0.05. Statistical analyses were performed using JMP version 14.1.0 (SAS institute, Cary, NC, USA).

## 3. Results

### 3.1. Demographic and Clinical Features of Women Based on the Fulfillment of Reproductive Desire

One hundred twenty-two women with HS were seen at our HS clinic between February 2017 and April 2020, with 85.2% of them (104/122) being of childbearing age: 49.04% (51/104) had fulfilled reproductive desires and 50.96% (53/104) had unfulfilled reproductive desires. Demographic and clinical features are shown in Table 1.

Regarding sociodemographic features, the women with fulfilled reproductive desires were older, and more frequently married or living with a partner. The proportion of patients with compulsory education was higher among women with unfulfilled reproductive desires and also in the proportion of employed women. BMI was higher among patients with fulfilled reproductive desires, as was the number of active smokers.

With respect to HS clinical features, the age of disease onset was lower in women with unfulfilled reproductive desires, but disease duration was longer in women with fulfilled reproductive desires. In concordance with longer disease duration, the Hurley stage, IHS4 and number of draining tunnels was higher in women with fulfilled reproductive desires. 

### 3.2. Treatment Patterns in HS Women of Childbearing Age

Global treatment patterns and their classification based on the fulfillment of reproductive desires are shown in Table 2. Pregnancy category (A B C D X), pregnancy impairments, recommended washout period before becoming pregnant and implications on pregnancy and fertility issues are also shown.

Intralesional corticosteroids were the most used treatment in both groups, followed by oral doxycycline, oral contraceptives and oral clindamycin. The prescription frequency was similar between groups except for combined oral contraceptives. They were more frequently prescribed in women with unfulfilled reproductive desires. 

Although non-statistically significant differences were found, the only treatment in the X category, acitretin, was prescribed less in woman with unfulfilled reproductive desires. Regarding category D treatments (doxycycline, levonorgestrel-releasing intrauterine system, spironolactone and systemic corticosteroids), the prescription rate was similar between groups. Biologics were used more in women with fulfilled reproductive desire 13.73% (7/51) vs. 3.77% (2/53), *p* = 0.08, both adalimumab and infliximab.

### 3.3. Demographic and Clinical Features of Women of Childbearing Age Receiving First-Line Biologic Treatment

Fourteen women started their first-line biologic treatment during follow-up at our HS clinic, Table 3. In total, 42.85% (6/14) of women had fulfilled reproductive desires, while 57.14% (8/14) had unfulfilled reproductive desires. Women with unfulfilled reproductive desires were younger than women with fulfilled reproductive desires, presenting an earlier disease onset.

All women with fulfilled reproductive desires were married or living with a partner, while in the group with unfulfilled reproductive desires, half were married, and half were separated. BMI, smoking habit, alcohol consumption and family history of HS were similar in both groups. 

In concordance with a higher disease duration severity evaluated by Hurley stage, the draining tunnel count was higher in women with fulfilled reproductive desires, while the abscess–nodule count was higher in women with unfulfilled reproductive desires. The mean IHS4 was similar in both groups. The number of affected areas and number of previous medical treatments was similar between groups. The number of previous surgical treatments was higher in woman with unfulfilled reproductive desires.

## 4. Discussion

In this study, we present the first clinical data describing the characteristics of HS women based on the accomplishment of their reproductive desires and the potential treatment impact on fertility, course and outcome of pregnancy.

Regarding sociodemographic features, women with unfulfilled reproductive desires were younger than women with fulfilled reproductive desires. The possibility of completing reproductive desires increases over the years [42], so an age difference between the two groups had been expected. The proportion of women with completed mandatory education level was higher in women with unfulfilled reproductive desires. This is explained by low educational level being associated with early maternal age [43], as a delay in the first childbirth may improve women’s educational and economic opportunities [44,45]. Higher BMI in women with fulfilled reproductive desires may be explained by their older age, as BMI is positively correlated with age [46], and the increasing weight gain between consecutive pregnancies, even in women without pre-existing obesity [47]. Weight gain is known to contribute to HS pathogenesis [48], while the effect of pregnancy on HS [49] as well as the effect of HS on pregnancy are unknown [50,51]. Although a high rate of HS exacerbation during pregnancy and postpartum has been reported, most patients did not receive HS-directed medical treatment or care from a dermatologist during their pregnancy [51]. Having a healthy life could also influence the rate of pregnancies. Having an unhealthy diet has been related to HS severity [18] and HS severity and polycystic ovary syndrome [19], factors that could decrease the possibility of being pregnant.

With respect to civil status, the lower proportion of patients married/living with partner in women with unfulfilled reproductive desire could not only be explained by age differences. When comparing these data to the civil status standardized by age in the Spanish population, greater differences were also found in the group with unfulfilled reproductive desires. The proportion of Spanish women who are married/living with partner in their 40s is nearly 65%, while in women in their 30s, this rate is around 40% [52]. Our results are in agreement with the Swedish HS Registry, which showed that HS patients are more frequently unmarried than the general population [53]. This may be explained by the fact that disease severity may make it difficult to find a partner [54] and have a negative impact on sex life [1,11]. Early disease onset may also have an influence on the difficulty of finding a partner as our results observed a higher frequency of single HS women in the early-onset group. 

Regarding clinical features, disease onset was earlier in women with unfulfilled reproductive desires. Disease duration was longer in women with fulfilled reproductive desires, as expected due to age difference. In concordance with a longer disease duration, disease severity assessed by Hurley stage and the number of draining tunnels were higher in women with fulfilled reproductive desires. Nevertheless, given the earlier age of onset in the women with unfulfilled reproductive desires, in the future, they could present more severe disease features with longer time of evolution than their counterparts with fulfilled reproductive desires. The higher number of draining tunnels in women with fulfilled reproductive desires is explained by their longer disease duration and their more severe disease. Although we did not find differences between affected areas in two groups, the involvement of the genitalia area might influence pregnancy desires, in agreement with higher psychological impact on these patients [55].

The decision regarding whether to have children is considered a major life-changing decision (MLCD), a contributor to the burden of chronic disease that could cause long-term effects on a patient’s life [54]. Experiences with their first pregnancy, genetic reasons, disease severity, complication worries, concerns about looking after their health and baby simultaneously, finances, the potential impact on family life and long-term treatment may influence the decision of whether or not to have more children, delay plans or have children through in vitro fertilization [54,56]. Getting married is also considered to be a MLCD [54], so chronic disease and disease severity could also influence this decision.

Concerning treatment patterns, almost no differences were found based on the fulfillment of reproductive desires. Combined oral contraceptives were more frequently prescribed in women with unfulfilled reproductive desire. They are a safe treatment and also play a role in avoiding undesired pregnancy [29]. Considering that unplanned pregnancies are very frequent [24,43] and deciding to have children is a complicated individual decision which has to be taken further in advance than among the general population, combined oral contraceptives may be an adequate treatment for HS patients of childbearing age. Although having to plan gestations could also mean a decrease in the birth rate, trying to ensure safety during pregnancy should be the overriding concern. Moreover, it would be important for clinicians to routinely ask HS women of reproductive age about their pregnancy intentions and advise those who are planning pregnancies what they can do to ensure optimal preconception to improve the health of both the woman and child and improve reproductive outcomes [57]. Many treatments prescribed are safe options for women of childbearing age. Only acitretin was in the X pregnancy category. Nevertheless, some frequently used medications are classified as D pregnancy category (doxycycline, resorcinol, spironolactone and systemic corticosteroids), so clinicians should take women’s reproductive desire into consideration. Regarding the rates of antibiotic prescription, they were high in both groups, doxycycline being the most frequent prescribed. It would be interesting to conduct targeted and specific antibiotic therapy as a high level of resistance has been described to antibiotics in HS patients, including rifampicin, clindamycin and tetracyclines [58].

Biologics were more used in women with fulfilled reproductive desire according to their more severe and longer disease, while anti-TNF agents seem to be safe before and during pregnancy [59]. Regarding features of the patients receiving first-line biologics, women with unfulfilled reproductive desires were younger and had an earlier onset of the disease, just like the general systemic treatment cohort. Although no differences in disease severity were found, because of the age difference, it is expected that if women with unfulfilled reproductive desires reach a similar age as women with fulfilled reproductive desires, they will have a severe condition. Hurley stage III was more frequent in women with fulfilled reproductive desires, while AN count was lower in this population. This is explained by the longer HS duration in this group being related to more tunnels and scars.

This study has some limitations: (1) The sample size is limited, although it was previously calculated and the sample was adjusted to the previous calculations; (2) the age difference between the two HS groups, inherent to the condition studied and the nature of a descriptive study. The aim of the study was to describe the clinical characteristics and treatment options for fertile women with HS according to their childbearing desires. To compare fertility rates, severity or other specific disease features, an age-matched study would be necessary and falls beyond the scope of this investigation. The strength of this study is that it provides a first overview regarding factors that could influence the accomplishment of reproductive desires in HS women and safety treatments during childbearing. Moreover, it was conducted by a dermatologist working in a specialized HS clinic with many years of experience treating HS patients.

## 5. Conclusions

In this study, we present the clinical and treatment profile of HS women of childbearing age based on the fulfillment of their reproductive desire. A higher educational level similar to that of the general population, and an earlier disease onset, with potential implications during teenage and early adulthood in finding a partner may make the fulfillment of reproductive desires difficult for patients with HS. Many treatments used for HS have a safe profile during childbearing age. Acitretin should be avoided, while anti-TNF alpha drugs may be a safe option. This study could help clinicians to achieve a better understanding of the specific characteristics of HS during childbearing age and consider reproductive desires when making treatment decisions.

## Figures and Tables

**Figure 1 life-11-00277-f001:**
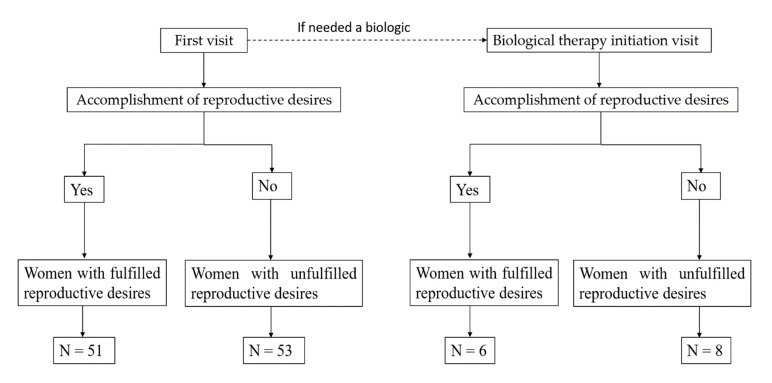
Flowchart of the participants.

**Table 1 life-11-00277-t001:** Demographic and clinical features of patients based on fulfillment of reproductive desires.

Variables	Women with Fulfilled Reproductive Desires (N = 51)	Women with Unfulfilled Reproductive Desires (N = 53)	*p*
**Age (years)**	42.06 (SD 6.46)	29.08 (SD 8.90)	<0.001 *
**Civil status** - **Married/living with partner** - **Separated/divorced** - **Single**	35 (69.63%) 2 (3.92%) 14 (27.45%)	12 (22.64%) 4 (7.55%) 37 (69.81%)	<0.001 *
**Mandatory education level (yes)**	30 (58.82%)	44 (83.02%)	0.007 *
**Unemployed (yes)**	26 (50.98%)	19 (35.85%)	0.120
**BMI (kg/cm^2^)**	32.64 (SD 7.61)	28.56 (SD 5.29)	0.002 *
**Smoking habit (yes)**	31 (60.78%)	23 (43.40%)	0.076
**Alcohol consumption (yes)**	9 (17.65%)	12 (22.64%)	0.526
**Age of onset (years)**	24.63 (SD 10.96)	18.23 (SD 7.16)	0.006 *
**Disease duration (years)**	17.43 (SD 9.29)	10.85 (SD 6.36)	<0.001 *
**Family history (yes)**	28 (54.90%)	27 (50.94%)	0.686
**Hurley stage** - **I** - **II** - **III**	13 (25.49%) 23 (45.10%) 15 (29.41%)	26 (49.06%) 22 (41.51%) 5 (9.43%)	0.010 *
**AN count**	1.86 (SD 2.07)	2.68 (SD 3.24)	0.130
**Draining tunnels count**	1.14 (SD 1.58)	0.49 (SD 0.80)	0.009 *
**IHS4**	7.33 (SD 6.75)	5.51 (SD 5.19)	0.124
**Number of affected areas**	1.90 (SD 1.06)	1.60 (SD 1.00)	0.145
**Number of previous medical treatments**	1.96 (SD 1.57)	1.68 (SD 1.36)	0.330
**Number of previous surgeries**	0.98 (SD 1.21)	1.04 (SD 1.81)	0.850

AN: total abscess and inflammatory nodule count; BMI: body mass index; IHS4: International Hidradenitis Suppurativa Severity Score System. Data are expressed as relative (absolute) frequencies and means (standard deviation (SD). The Student’s *t* test for independent samples was used to compare continuous variables and the chi-square test or Fisher’s exact test, as appropriate, were applied to compare categoric data. Two-tailed * *p* < 0.05 was considered statistically significant in all tests.

**Table 2 life-11-00277-t002:** Treatments of the sample based on fulfillment of reproductive desires, pregnancy category, washing time recommendation to get pregnant, possible impact on pregnancy and fertility.

Treatment	Total (N = 104)	Women with Fulfilled Reproductive Desires (N = 51)	Women with Unfulfilled Reproductive Desires (N = 53)	*p*	Pregnancy Category ABCDX	Does It Make Pregnancy Impossible?	Recommended Washing Time to Get Pregnant (Weeks)	Fertility Comments
**Intralesional Corticosteroids** [22,27]	59 (56.73%)	28 (54.90%)	31 (58.49%)	0.136	C	No. Safe in pregnancy	Safe in pregnancy	Not reported in women
**Doxycycline** [28]	36 (34.62%)	16 (31.3%)	20 (37.7%)	0.49	D	Yes. Teratogenic, risk of dental staining, poor bone growth.	1 ^#^	Unknown. Possible risk of contraceptive failure
**Oral contraceptives** [29] **- Combined oral contraceptives** **- Desogestrel**	33 (31.73%)	13 (25.49)	20 (37.74%)	0.10	B	No. Little or no increased risk of birth defects in women who inadvertently use during early pregnancy	2 ^#^	Reversible loss of fertility
	21 (20.19%)	5 (9.80%)	16 (30.19%)	0.013 *				
	12 (11.54%)	8 (15.69%)	4 (7.55%)	0.231				
**Oral clindamycin** [30]	15 (14.42%)	10 (19.61%)	5 (9.43%)	0.140	B	No. Not contraindicated during pregnancy, no evidence of teratogenicity	Safe in pregnancy	No effects on fertility or mating ability reported.
**Resorcinol** [22,31]	8 (7.69%)	4 (7.84%)	4 (7.55%)	1.00	D	No. Insufficient safety data	One day ^#^	May not affect the reproductive performance and fertility
**Adalimumab** [32]	8 (7.69%)	6 (11.76%)	2 (3.77%)	0.156	B	No. Safety unclear no increased risk of adverse birth outcomes to date	24	No fertility rate reported to date
**Levonorgestrel-releasing intrauterine system** [33]	6 (5.77%)	4 (7.84%)	2 (3.77%)	0.432	D	Yes. Ectopic pregnancy, pregnancy loss, septic abortion	1 ^#^	Reversible loss of fertility
**Surgery** [22,34]	5 (4.81%)	2 (3.92%)	3 (5.66%)	0.679	B	No. Lidocaine epinephrine is local anesthetic of choice. Procedures requiring general anesthesia should be deferred	One day	No effect
**Spironolactone** [35]	4 (3.85%)	3 (5.88%)	1 (1.89%)	0.358	D	No. May affect sex differentiation for the male during embryogenesis due to anti-androgenic properties	1 ^#^	May impair mating, fertility, and fecundity
**Acitretin** [36]	4 (3.85%)	3 (5.88%)	1 (1.89%)	0.358	X	Yes. Embryotoxic and/or teratogenic	144	Reversible mild to moderate spermatogenic/Not fertility impairment reported in women
**Colchicine** [37]	4 (3.85%)	3 (5.88%)	1 (1.89%)	0.358	C	No. Risk of teratogenicity is discussed.	One day ^#^	Not established
**Rifampin** [38]	3 (2.88%)	2 (3.92%)	1 (1.89%)	0.614	C	No. Insufficient data. May increase the risk for maternal postpartum hemorrhage and bleeding in the exposed infant when administered during the last few weeks of pregnancy	1 ^#^	Fertility not affected. Possible risk of contraceptive failure concomitant with combined oral contraceptives
**Metformin** [39]	2 (1.92%)	0 (0.00%)	2 (3.77%)	0.495	B	No	Safe in pregnancy	No effect
**Systemic Corticosteroids** [22,40]	2 (1.92%)	1 (1.96%)	1 (1.89%)	1.00	D	No. Fetal harm can occur with first trimester use. Small risk of oral cleft deformity, no risk of major anomalies	One day ^#^	Not formally evaluated Menstrual irregularities.
**Infliximab** [41]	1 (0.96%)	1 (1.96%)	0 (0.00%)	0.49	B	No. Safety unclear no increased risk of adverse birth outcomes to date	24	Unknown

^#^ If no specified information was found, safety was considered after six half-lives, when 98% of a drug is eliminated from the body. Data are expressed as relative (absolute) frequencies. The chi-square test or Fisher’s exact test, as appropriate, were applied to compare categoric data. Two-tailed * *p* < 0.05 was considered statistically significant in all tests.

**Table 3 life-11-00277-t003:** Demographic and clinical features of patients receiving first-line biologic treatment based on fulfillment of reproductive desires.

Variable	Women with Fulfilled Reproductive Desires (N = 6)	Women with Unfulfilled Reproductive Desires (N = 8)	*p*
**Age (years)**	46.50 (SD 4.28)	30.13 (SD 11.06)	0.005 *
**Civil status** - **Married/living with partner** - **Separated/divorced** - **Single**	6 (100%) 0 0	4 (50.00%) 0 4 (50.00%)	0.084
**Educational level** - **Mandatory**	6 (100.00%)	8 (100.00%)	1
**BMI (kg/cm^2^)**	34.39 (SD 5.72)	33.11 (SD 6.68)	0.71
**Smoking habit (yes)**	3 (50.0%)	3 (37.50%)	0.64
**Alcohol consumption (yes)**	1 (16.7%)	3 (37.50%)	0.804
**Family history (yes)**	4 (66.67%)	4 (50.00%)	0.627
**Age of onset (years)**	26.83 (SD 9.41)	15.50 (SD 5.73)	0.016 *
**Disease duration (years)**	19.67 (SD 9.58)	14.63 (SD 5.49)	0.38
**Hurley stage** - **I** - **II** - **III**	0 (0.00%) 1 (16.67%) 5 (83.33%)	2 (25.00%) 2 (25.00%) 4 (50.00%)	0.332
**AN count**	1.83 (SD 1.72)	4.13 (SD 5.49)	0.347
**Draining tunnels count**	2.60 (1.75)	2.00 (SD 1.77)	0.497
**IHS4**	13.83 (SD 6.88)	14.38 (SD 11.51)	0.921
**Number of affected areas**	3.33 (SD 1.86)	3.13 (SD 1.13)	0.799
**Number of previous medical treatments**	4.50 (SD 2.74)	4.25 (SD 1.28)	0.822
**Number of previous surgeries**	1.00 (SD 0.89)	2.75 (SD 2.25)	0.099

AN: total abscess and inflammatory nodule count; BMI: body mass index; IHS4: International Hidradenitis Suppurativa Severity Score System. Data are expressed as relative (absolute) frequencies and means (standard deviation (SD). The Student’s *t* test for independent samples was used to compare continuous variables, and the chi-square test or Fisher’s exact test, as appropriate, were applied to compare categoric data. Two-tailed * *p* < 0.05 was considered statistically significant in all tests.

## Data Availability

The data presented in this study are available on request from the corresponding author.

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
