# Peer review of "Reproductive Potential and Outcomes in Patients with Hidradenitis Suppurativa: Clinical Profile and Therapeutic Implications"

_life, 2021, doi:10.3390/life11040277_

Round 1
Reviewer 1 Report
The results of this study showed that the drainage tunnels count was significantly higher in women with fulfilled reproductive desires, while the abscess and nodule count was higher in women with unfulfilled reproductive desires. What could explain these significant differences in clinical manifestations? This should be discussed in the Discussion Section
Author Response
The results of this study showed that the drainage tunnels count was significantly higher in women with fulfilled reproductive desires, while the abscess and nodule count was higher in women with unfulfilled reproductive desires. What could explain these significant differences in clinical manifestations? This should be discussed in the Discussion Section.
Thank you four your suggestions. Hidradenitis suppurativa is a chronic disease characterized by painful nodules, abscesses, and draining tunnels. Draining tunnels are permanent lesion that can only disappear after surgery treatment. So, they are more common in patients with severe disease and long disease duration. As women with fulfilled reproductive desires are older and have a more severe disease, a higher prevalence of draining tunnels is also expected. This information has been added to the discussion section.
Reviewer 2 Report
This is a very interesting study, which explores a deep area of HS patient life. I have some notes:
1) Please report EC number approval
2) Please describe how the diagnosis of HS was performed.
3) It would be interesting to analyse if the desire of having a baby of HS women is influenced by the involvment of the genital and inguinal area by HS lesions. Try to argument the psychological impact of inguinal and genital HS on the reproductive desire, if any correlation is found.
4) When describing antibiotic therapy, it would be useful to mention the problem of bacterial antibiotic resistance
Author Response
This study was approved by the Ethics Committee of the Hospital Universitario Virgen de las Nieves, the 22nd of December 2019, reference number 2390-N-19, and is in accordance with the Helsinki Declaration. This is a very interesting study, which explores a deep area of HS patient life. I have some notes:
Thank you for your suggestions.
1) Please report EC number approval
This study was approved by the Ethics Committee of the Hospital Universitario Virgen de las Nieves, the 22nd of December 2019, reference number 2390-N-19, and is in accordance with the Helsinki Declaration. This information has been added to the material and methods section.
2) Please describe how the diagnosis of HS was performed.
The diagnosis of HS was performed by a dermatologist following the criteria recorded in the HS clinical guidelines (Alikhan et al. 2019): Diagnosis relies on clinical findings of (1) typical HS lesions, (2) predilection for intertriginous sites, and (3) recurrence. This information has been added to the material and methods section.
3) It would be interesting to analyse if the desire of having a baby of HS women is influenced by the involvment of the genital and inguinal area by HS lesions. Try to argument the psychological impact of inguinal and genital HS on the reproductive desire, if any correlation is found.
We did not find differences regarding the are involved. Anyway, a comment regarding the psychological impact of inguinal and genital HS on the reproductive desire has been added in the discussion section.
4) When describing antibiotic therapy, it would be useful to mention the problem of bacterial antibiotic resistance
Information regarding this topic has been added in the discussion. The rates of antibiotic prescription were high in both groups, being doxycycline the most frequent prescribed was. It would be interesting to conduct targeted and specific antibiotic therapy as it has been described high level of resistance to antibiotics in HS patients, including rifampicin, clindamycin and tetracyclines.
Reviewer 3 Report
Trinidad Montero-Vilchez et al. reported the clinical and treatment profile of HS women of childbearing age based on the fulfillment of their reproductive desire.
Here are my suggestions.
Abstract
Pg 1, line 21: IHS4 ?
Pg 1, lines 22-23: “desires (30.19%vs9.80%) while biologics were less used in this group (3.77%vs13.73%)”. Please add significance
Introduction
The introduction ​appears way too superficial in the description of the rationale, and lacks important pieces of information and data, which would support and emphasize the importance of the current study. Some studies need to be reported and illustrated.
Authors should include the role of diet affecting both hidradenitis (See PMID: 30597889) and reproductive function (See PMID: 28521685), as reported by studies in patients with polycystic ovary syndrome (See PMID: 31547562).
Pg 2, line 50: “diabetes” please add “type 2”
METHODS
Demographic and clinical characteristics of patients were recorded. Which? How?
Participants
The protocol number of local Ethics Committee is missing.
Was the study conducted without sponsorship?
The inclusion / exclusion criteria should be better described.
During which phase of the menstrual cycle were women evaluated?
How long had hidradenitis been diagnosed?
Are there data on the dietary habits of the participants?
Accomplishment of reproductive desires. Was a questionnaire used? Which?
How were the following variables evaluated? Civil status, level of education, smoking habit, and alcohol intake.
Has physical activity been assessed? How?
Clinical features included age at HS onset, disease duration, Hurley stage, number of affected areas, previous medical and surgical treatments, nodules, abscesses and draining tunnels count.
Who did these assessments? A dermatologist?
Furthermore:
A flow chart could be appropriate to demonstrate how participants were included in the study.
Statistical analysis
Statistical analysis should explain the single test used for each variable.
The authors should calculate the sample size.
Results
Authors should perform partial correlation for confounding variables.
Discussion
The study has several limitations that were not addressed by the authors. It would be helpful to add the strengths as well.
The limited sample size represents a strong limitation therefore I suggest adding a calculation of the statistical power of the data and the sample size.
Furthermore, both discussion and conclusions need to be expanded and the relevance and contribution for the scientific community of the results reported should be clarified and explained.
Author Response
Trinidad Montero-Vilchez et al. reported the clinical and treatment profile of HS women of childbearing age based on the fulfillment of their reproductive desire.
Here are my suggestions.
Abstract
Pg 1, line 21: IHS4 ?
IHS4 means International Hidradenitis Suppurativa Severity Score System. It has been added. The meaning has been added.
Pg 1, lines 22-23: “desires (30.19%vs9.80%) while biologics were less used in this group (3.77%vs13.73%)”. Please add significance
Significance has been added.
Introduction
The introduction ​appears way too superficial in the description of the rationale, and lacks important pieces of information and data, which would support and emphasize the importance of the current study. Some studies need to be reported and illustrated.
Authors should include the role of diet affecting both hidradenitis (See PMID: 30597889) and reproductive function (See PMID: 28521685), as reported by studies in patients with polycystic ovary syndrome (See PMID: 31547562).
The introduction has been extended and all the references recommended has been added.
Pg 2, line 50: “diabetes” please add “type 2”
Type 2 has been added
METHODS
Demographic and clinical characteristics of patients were recorded. Which? How?
Demographic and clinical characteristics were recorded by means of clinical interview and physical examination. Demographic characteristic included sex, age, civil status, level of education, body mass index (BMI), smoking habit, alcohol consumption and family history of HS. Clinical features included age at HS onset, disease duration, Hurley stage, number of affected areas, previous medical and surgical treatments, nodules, abscesses and draining tunnels count. It was added in the text.
Participants
The protocol number of local Ethics Committee is missing.
This study was approved by the Ethics Committee of the Hospital Universitario Virgen de las Nieves, the 22nd of December 2019, reference number 2390-N-19. This information has been added.
Was the study conducted without sponsorship?
No, it wasn’t. This information is included in the funding section. This research received no external funding.
The inclusion / exclusion criteria should be better described.
They had been better described.
During which phase of the menstrual cycle were women evaluated?
The phase of the menstrual cycle when the women were evaluated was not recorded.
How long had hidradenitis been diagnosed?
Women with fulfilled reproductive desires have been diagnosed from HS for more than 17 years and women with unfulfilled reproductive desires have been diagnosed from HS for more than 10 years. This information ca be found in table 1.
Are there data on the dietary habits of the participants?
Thank you for your suggestion. Dietary habits were not investigated.
Accomplishment of reproductive desires. Was a questionnaire used? Which?
To evaluate the accomplishment of reproductive desires, women were asked if they wanted to have a baby now or in the future during the visit and they were gathered into two groups. If the answer was not, women were classified in the group of women with fulfilled reproductive desires; while if they answered yes, they were included in the group of women with unfulfilled reproductive desires. No questionnaire was used.
How were the following variables evaluated? Civil status, level of education, smoking habit, and alcohol intake.
Civil status, level of education, smoking habit, and alcohol intake were recorded by means of clinical interview. This information was added in the text
Has physical activity been assessed? How?
Thank you for your suggestion. Physical activity was was not investigated.
Clinical features included age at HS onset, disease duration, Hurley stage, number of affected areas, previous medical and surgical treatments, nodules, abscesses and draining tunnels count.
Who did these assessments? A dermatologist?
They were evaluated by physical examination that was carried out by a dermatologist.
Furthermore:
A flow chart could be appropriate to demonstrate how participants were included in the study.
A flow chart to demonstrate how participants were included in the study has been provided.
Statistical analysis
Statistical analysis should explain the single test used for each variable.
The student’s t-test or the Wilcoxon-Mann-Whitney test were used to compare continuous data (such as age or AN count), and the χ2 test or Fisher’s exact test were applied to nominal data where necessary (such as civil status or Hurley stage). To avoid repetition an example of each variable is provided. In the foot of the table the test are also mentioned.
The authors should calculate the sample size.
The sample size has been provided.
Results
Authors should perform partial correlation for confounding variables.
The aim of the study is to explore sociodemographic features, clinical characteristic and treatment prescribed between women with unfulfilled and fulfilled reproductive desires. This is an exploratory study so we believe that adjusted analyses are not necessary. If necessary, correlation between variables could be carried out.
Discussion
The study has several limitations that were not addressed by the authors. It would be helpful to add the strengths as well.
We have added the strengths in the discussion. If you consider our study have other limitations that are not provided in the discussion, please let us know so we are able to add them in the discussion.
The limited sample size represents a strong limitation therefore I suggest adding a calculation of the statistical power of the data and the sample size.
We have added in the material and method section, how was the sample size calculated.
Furthermore, both discussion and conclusions need to be expanded and the relevance and contribution for the scientific community of the results reported should be clarified and explained.
The discussion and the conclusions have been expanded. We believe that articles helping to understand specific conditions during childbearing age and helping clinicians to make treatment decision considering reproductive desires are really important, even more in European countries where the low birth rate is a major healthy, political, social and economic problem.
Round 2
Reviewer 1 Report
-
Author Response
Changes were made.
Thank you for the comments
Reviewer 3 Report
The authors have successfully addressed the issues raised in this article. I recommend to accept this article in Life Journal.
Author Response

(The authors gave the same response as above.)
